# West Nile virus transmission potential in Portugal

José Lourenço [1,2✉], Sílvia C. Barros[3], Líbia Zé-Zé [2,4], Daniel S. C. Damineli [5], Marta Giovanetti [6,7], Hugo C. Osório[4,8], Fátima Amaro[4,8], Ana M. Henriques[3], Fernanda Ramos[3], Tiago Luís[3], Margarida D. Duarte[3], Teresa Fagulha[3], Maria J. Alves[4,8] & Uri Obolski [9,10]

It is unclear whether West Nile virus (WNV) circulates endemically in Portugal. Despite the country's adequate climate for transmission, Portugal has only reported four human WNV infections so far. We performed a review of WNV-related data (1966–2020), explored mosquito (2016–2019) and land type distributions (1992–2019), and used climate data (1981–2019) to estimate WNV transmission suitability in Portugal. Serological and molecular evidence of WNV circulation from animals and vectors was largely restricted to the south. Land type and climate-driven transmission suitability distributions, but not the distribution of WNV-capable vectors, were compatible with the North-South divide present in serological and molecular evidence of WNV circulation. Our study offers a comprehensive, data-informed perspective and review on the past epidemiology, surveillance and climate-driven transmission suitability of WNV in Portugal, highlighting the south as a subregion of importance. Given the recent WNV outbreaks across Europe, our results support a timely change towards local, active surveillance.

[1] Department of Zoology, University of Oxford, Oxford, United Kingdom. [2] Biosystems and Integrative Sciences Institute, Edificio TecLabs, Campus da FCUL, Lisboa, Portugal. [3] Instituto Nacional de Investigação Agrária e Veterinária, Virology Laboratory, Oeiras, Portugal. [4] Centro de Estudos de Vectores e Doenças Infecciosas, Instituto Nacional de Saúde Doutor Ricardo Jorge, Marateca, Portugal. [5] Department of Pediatrics, Faculdade de Medicina da Universidade de São Paulo, São Paulo, Brazil. [6] Laboratório de Flavivírus, Instituto Oswaldo Cruz Fiocruz, Rio de Janeiro, Brazil. [7] Laboratório de Genética Celular e Molecular, Universidade Federal de Minas Gerais, Minas Gerais, Brazil. [8] Instituto de Saúde Ambiental, Faculdade de Medicina da Universidade de Lisboa, Lisboa, Portugal. [9] School of Public Health, Faculty of Medicine, Tel Aviv University, Tel Aviv, Israel. [10] Porter School of the Environment and Earth Sciences, Faculty of Exact Sciences, Tel Aviv University, Tel Aviv, Israel. ✉email: jose.lourenco@zoo.ox.ac.uk

West Nile virus (WNV) is an RNA mosquito-borne virus of the Flaviviridae family, first identified in 1937 in the West Nile district of Uganda. WNV ecology is characterised by a zoonotic transmission cycle maintained between mosquitoes and wild avian species. Six mosquito genera have been implicated in WNV transmission (*Anopheles, Aedes, Culex, Culiseta, Mansonia, Ochlerotatus*), but *Culex spp.* (e.g., *Cx. pipiens, Cx. univitattus/perexiguus, Cx. modestus*) are commonly referred to as the main vectors of WNV in Europe and North America. Occasional viral spillover can occur from this cycle to humans as well as domesticated and wild mammals[1–3]. From such incidental hosts, humans and equines feature predominantly in epidemiological data given a clear West Nile fever clinical profile, with equines often serving as a sentinel species in many countries. Contrary to most avian species, mammals are inefficient amplifier hosts, and hence can not establish mosquito-mammal transmission cycles[3–5]. Approximately 80% of infections in humans are asymptomatic while the rest may develop mild or severe disease of neuroinvasive nature and potentially death[6]. There are currently no licensed vaccines nor antiviral treatments for humans[3,7], but four licensed vaccines are on the market for use in equines[7].

Global trends in climate change and human factors continue to favour the dispersal of mosquito species into new regions, and the long-distance movement of infectious hosts across the globe. Both factors promote the introduction and epidemic activity of mosquito-borne viruses such as WNV into previously unaffected areas[8–11]. Some of the most relevant mosquito-borne examples of the past 10 years include the introduction of chikungunya and Zika viruses into the Americas[12,13], as well as the resurgence of yellow fever virus in Brazil and regions of Africa[14,15].

In the 20th century, WNV was mostly reported to circulate in Israel and some African countries[16]. More recently, in 1999, it gained international notoriety after its introduction into the United States of America (USA, New York), from where it quickly became endemic in most of the USA and Canada[17]. Concurrently and in the following two decades, epidemic activity increased in regions of Europe, the Middle East, and Russia[11,16]. Presently, WNV epidemiology shows large geo-temporal heterogeneities across countries and continents, but much of that variation is not well understood. However, given its well documented effects on the life-cycles and dispersal of avian and mosquito species, climate has been acknowledged as a major factor influencing local WNV epidemic activity, dispersal, and persistence[3,18,19].

The recent epidemiological history of WNV in Europe is characterised by increasing epidemic activity in countries including France, Italy, Greece, Hungary, Romania, Bulgaria, Serbia and Ukraine[20,21]. In 2018, the continent reported its largest epidemic ever across many countries, exceeding cumulative infections since 2010[22]. In 2020, the first autochthonous human infections were reported in the Netherlands[23], and Spain had its largest epidemic to date including regions previously not affected[24,25]. Compared to Spain and other countries in the Mediterranean basin, Portugal remains an outlier in so far having reported only four, serologically confirmed human infections in the past 40 years.

Routine laboratory diagnostics in Portugal were established in 1996, and are currently managed by the Centro de Estudos de Vectores e Doenças Infecciosas (CEVDI, Center for Vectors and Infectious Diseases Research) of the Instituto Nacional de Saúde Dr. Ricardo Jorge (INSA, National Health Institute Dr. Ricardo Jorge) and by the Instituto Nacional de Investigação Agrária e Veterinária (National Agrarian and Veterinary Research Institute, INIAV) in human and animals, respectively. Notification of WNV-related disease is currently mandatory in Portugal. Being a European Union (EU) member state, Portugal reports outbreaks of equine WNV neurological disorders to the European Animal Disease Notification System[26], and annual reports for the EU region are published by the European Food Safety Authority[27] and European Centre for Disease Prevention and Control (ECDC)[28].

Since 2008 the country has been performing extensive mosquito surveillance under REVIVE (REde de VIgilância de VEtores, surveillance network for vectors[29,30]). A multitude of national agencies within the Ministry of Health are involved, including the General Directorate of Health (DGS), INSA, the five Regional Health Administrations (Algarve, Alentejo, Lisboa e Vale do Tejo, Centro, and Norte), and the non-continental agencies the Institute of Health Administration of Madeira and the Regional Health Directorate of the Azores. REVIVE is responsible for the nationwide surveillance of the most significant hematophagous arthropods for national public health (mosquitoes, ticks, and sandflies) including the cataloguing of extant species and their population sizes. Surveillance of invasive mosquito species, such as *Aedes aegypti* and *Ae. albopictus*, and screening (e.g. using PCR) of a multitude of field-collected mosquito-species for viruses is also regularly performed. According to International Health Regulations, airports, ports, storage areas, and specific border regions with Spain are regularly monitored throughout the year according to the needs of local and regional authorities. Reports from REVIVE are provided regularly to national agencies across the country[29], and to the European Network for Arthropod Vector Surveillance for Human Public Health, a network for data sharing on the geographic distribution of arthropod vectors, transmitting human and animal disease agents (originally known as VBORNET, and VectorNet since 2014[31]).

WNV surveillance in Portugal remains passive and mostly reactive to occasional reports of clinically compatible events in equines and humans. Furthermore, according to the experience of the national reference laboratory for vector-borne viruses, practitioners in Portugal lack awareness of the importance of WNV screening in compatible disease cases. As such, and since infections tend to be asymptomatic or mild, it is possible that only a very small percentage of all infections end up being screened and therefore accounted for. Detection of WNV RNA in serum, cerebrospinal fluid, or urine offer significant technical challenges in both humans and equines when assessed substantially after the start of the symptomatic period. In Portugal, WNV diagnosis in humans is typically requested following the failure of all other disease-compatible diagnoses, commonly days or weeks after symptoms onset and consequently after the viraemic phase. The reactive nature of the surveillance system also implies that increased awareness and screening requests for humans and equines tend to rise around areas with recent positive cases, but only for very short time windows which remain insufficient for long-term risk and epidemiological assessment. These aforementioned realities allow for systematic gaps in current knowledge regarding the local epidemiology of WNV, hindering important conclusions such as whether or where the virus is endemic.

In this study, we review and describe the international and Portuguese literature and official data sources (1966–2020), explore mosquito data (2016–2019) and land type distribution (1992–2019) across the country, further providing a theoretical geo-temporal assessment of WNV transmission potential in Portugal (1981–2019). We find a clear North-South dichotomy in existing evidence for the circulation of WNV in Portugal, although high-resolution data on WNV-capable mosquito species suggests transmission would be possible across the entire country. Theoretical assessments of WNV transmission potential in

Portugal based on climate data suggest that the south of the country presents climate-driven seasonal signatures that justify the limited evidence of WNV circulation being restricted to that region. Using climate data of the past four decades, we also quantify trends in climate and their possible historical effects on the local WNV transmission potential.

## Results

**Evidence for West Nile virus circulation in Portugal**. The earliest evidence of WNV circulation in Portugal arose from serological surveys in the south of the country in 1966–1967, reporting high titres of hemagglutination-inhibiting antibodies and neutralizing capacity against WNV in animal serum[32,33]. Such findings led to several initiatives in 1969 to create institutional links between the national health and veterinary medicine services with the purpose of seeding future studies and raising awareness to the virus[34]. Some of the earlier initiatives included the capture and testing of animals (e.g. equines, rodents, birds, mosquitoes) and serological testing of human serum[33–36], eventually leading to the first viral isolation from the mosquito species *Anopheles maculipennis*[35].

After three decades with seemingly no relevant research output in the country, further evidence for the circulation of WNV in Portugal accrued from several studies. Between 1999 and 2002, Formosinho and colleagues reported serological data from several avian species sampled across natural reserves and from equines sampled in the district of Santarém[37]. A few studies in the period 2004-2010 also reported positive findings in several avian species[38], mosquitoes[39,40], and equines[38]. More recently, between 2015 and 2020, following reports of several clinically compatible equine outbreaks in the south of the country, a number of equines were found serologically positive to the virus in the Alentejo and Algarve[41,42]. Some of this evidence is reported for the first time in this study (see **Methods** for details, and Supplementary Table 1 for summary).

Although such evidence supports a long history of local circulation in animals and mosquitoes, autochthonous WNV infections in humans in Portugal were only first reported in 2004. This was the case of two Irish bird-watchers found positive for both WNV IgG and IgM antibodies. They visited Ria Formosa, a marshland rich in endemic and migratory birds in the Algarve between the 26th June and 10th of July[43]. Two studies published thereafter were able to detect WNV by RT-PCR in pools of mosquitoes collected near the putative region of infection that year[39,40]. A third case was reported in July 2010 in the region of Setúbal, from a local rural resident with no history of international travel[44]. The resident tested positive for both WNV IgG and IgM antibodies in the weeks following the symptoms onset. Reactive mosquito collection performed around the patient's residence between the 13 and 18th of July revealed species suited for WNV transmission (e.g. *Cx. pipiens*), although all mosquito pools tested negative by RT-PCR[44]. Two months later, two equine cases were notified about 4 kilometers away from the reported human case. The most recent human infection was reported in a rural area of Almancil (Algarve) in 2015 in a patient testing positive for both WNV IgG and IgM antibodies and viral neutralization, with no record of international travel or history of vaccination against flaviviruses[45]. In August and September of the same year several outbreaks in equines were reported in Loulé (Algarve) with some animals testing positive by serology[42].

It is likely that the main reason why none of the human or equine reported infections in Portugal have so far been confirmed by RT-PCR is due to diagnosis and screening being performed well past the viraemic phase. This might also explain why

obtaining sequencing data related to WNV Portuguese infections has not been successful. To date, a single study has been published on the local phylogenetics of WNV, with sequencing solely obtained from mosquito pools (Supplementary Table 3)[40]. Such sequences, based on a short segment of the NS5 gene, demonstrated a close genetic relationship with past WNV lineages circulating in the Mediterranean basin (France, Morocco, and Italy), but fell short in providing definite conclusions on viral diversity, phylodynamics, and persistence in the country.

When summarized, the past evidence for the circulation of WNV in Portugal is mostly restricted to reports from the south of the country, in particular the districts of Faro, Beja, Setúbal, Évora, Lisboa, Portalegre, and Santarém (Fig. 1, Supplementary Table 1). Such reports include evidence from avian species (e.g. *Accipiter gentilis, Ciconia ciconia, Strix aluco*), equines, mosquitoes (e.g. *Cx. pipiens, Cx. univittatus*) and humans, achieved via different confirmation methods. It is unknown whether this North-South dichotomy in observed WNV circulation stems from sampling bias or from disparate ecological conditions dictating different WNV epidemic activity.

On one hand, sampling bias could be due to large differences in regional biodiversity of host species (e.g. there is evidence that the equine population is larger in the south[46]), or a propensity to focus on the south for other academic and surveillance reasons (e.g. where human and equine cases have been reported historically). On the other hand, disparate ecological conditions could be dictated by factors such as local differences in climate and landscape.

**West Nile virus vectors in Portugal**. To date, very few articles in the research literature have provided geo-temporal data on the population size and dynamics of mosquito species relevant for the transmission of WNV in Portugal (Supplementary Table 2). To explore possible determinants of the observed North-South dichotomy in WNV evidence, we obtained REVIVE *Cx. pipiens* data collected between 2017 and 2019 across Portuguese counties (Fig. 2). The methods of collection (e.g. type of trap used), as well as resources and regional objectives, varied by year, as described in detail in the published reports[29]. Summarizing the data revealed that both sampling coverage and the total number of yearly collections per county varied in time, both for adult (Fig. 2a–c) and immature (Fig. 2e–g) forms of this mosquito species. We noticed that even in the presence of many project-related factors affecting sampling, *Cx. pipiens* was found to be present across the country (Fig. 2d, h). Since *Cx. pipiens* is considered the main vector for transmission of WNV in Portugal[47], these findings were in contrast with the North-South divide of WNV circulation found in the data review from the literature and public databases (Fig. 1, Supplementary Table 1).

**Land types in Portugal**. The distribution of land types (e.g. cropland, urban, natural) can modulate the transmission of mosquito-borne pathogens and the risk for cross-species transmission[48,49]. Therefore we used the distribution of different land types in Portugal (see **Land cover and land types** for details) to explore how they could contribute to the explanation of the North-South dichotomy in WNV circulation data.

We mapped the spatial distribution of land types (Fig. 3a) and their frequencies between 1992 and 2019 (Fig. 3b). The percentage of mixed land (natural, cropland) was similar between the regions (~23% of all land). The south was enriched in cropland (~40%), approximately twice as much as in the north. In contrast, the north was dominated by natural vegetation (~55%), approximately double what was observed in the south. The spatial distribution of different land types, which was quite stable over

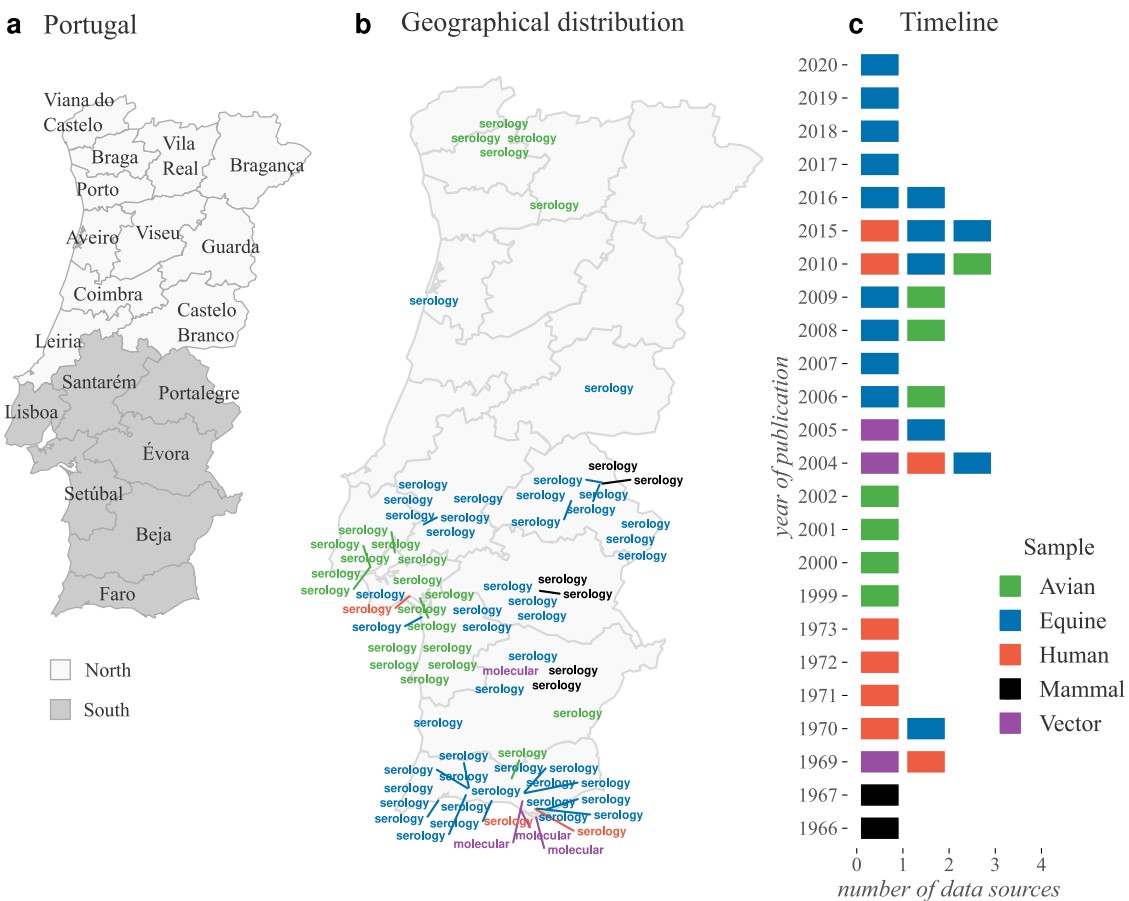

**Fig. 1 Past evidence of WNV circulation in Portugal.** Summary of data sources (articles, national reports) on various types of evidence for the circulation of WNV in Portugal. **a** presents the official (district) divisions of Portugal, colored according to north and south areas. **b** presents the geographical region of each data source (latitude-longitude). Positioning is only approximate, given that some reports refer to regions of different areas. For data sources referring to large areas (e.g. districts), the capital city of the region was used. Not all sources are mapped, e.g. not reporting where positive samples were found. Mapped words represent the type of evidence reported, aggregated into two categories: molecular, for PCR, isolation; and serology, for HI, VNT, ELISA. **c** shows a timeline of the number of data sources per year. The color scale (on the right) represents the different types of samples reported, and refers to both panels **b** and **c**. Not all years between 1966 and 2020 had data sources. A detailed summary of data sources in Supplementary Table 1.

time, highlighted differences between the regions that were reminiscent of the North-South divide in existing WNV data. The abundance of cropland in the south appeared linked with reports of WNV circulation, which conforms to mounting evidence of the positive link between cropland and the transmission activity of WNV both in humans and animals[49–53].

**Climate-driven West Nile virus transmission potential in Portugal.** Climate is a major driver of the population dynamics of mosquito and avian species participating in the zoonotic cycle of WNV[49]. According to the Köppen climate classification, Portugal presents a North-South divide in climate types, with the northern region classified as Temperate Mediterranean, and the southern region as the Warm Mediterranean (see Supplementary Fig. 2 for a spatial distribution summary of climatic variables). We used a climate-driven suitability index (termed index P) to estimate the geo-temporal transmission potential of WNV in Portugal. We aimed to understand whether the two types of climate could explain the North-South divide in existing WNV data, and to gain insights on the possible historical effects of long-term climate trends.

This index estimates the transmission potential of each adult female mosquito in the zoonotic reservoir, and can be interpreted as the risk of spillover to the human and equine populations (see **Transmission Suitability** for details). Typically, climate-driven suitability indices are validated by quantifying geo-temporal

correlations between time series of the index and suspected or confirmed infections. However, this was not possible due to the absence of reported chains of transmission in Portugal. Nevertheless, the index was previously validated for WNV in Israel[54], and for other mosquito-borne viruses elsewhere[13,55,56]. In this context, we proceeded to compare our estimated transmission potential with empiric mosquito data.

We thus first compared the only available time-series data related to WNV transmission in Portugal, which included the newly estimated index P and regional *Cx. pipiens* population sizes from the REVIVE network (Fig. 4). Given the North-South disparity in past evidence for the circulation of WNV (Fig. 1) and the widespread geographical distribution of *Cx. pipiens* (Fig. 2), we compared directly two of the most northern and southern districts of the country. Although the index P measures transmission per mosquito and not population size, we found the index to present peaks and troughs coinciding in time with those of the mosquito population sizes, both in the south (Fig. 4a, c) and north (Fig. 4b, d). Apart from the previously demonstrated potential for the index P to be correlated with WNV human infections[54], these results thus suggest that (at least in Portugal) the index can also be used to estimate the seasonal timing of variation in *Cx. pipiens* population sizes.

The patterns in Fig. 4a–d further suggested several characteristics of the typical season of WNV transmission in Portugal, as informed by

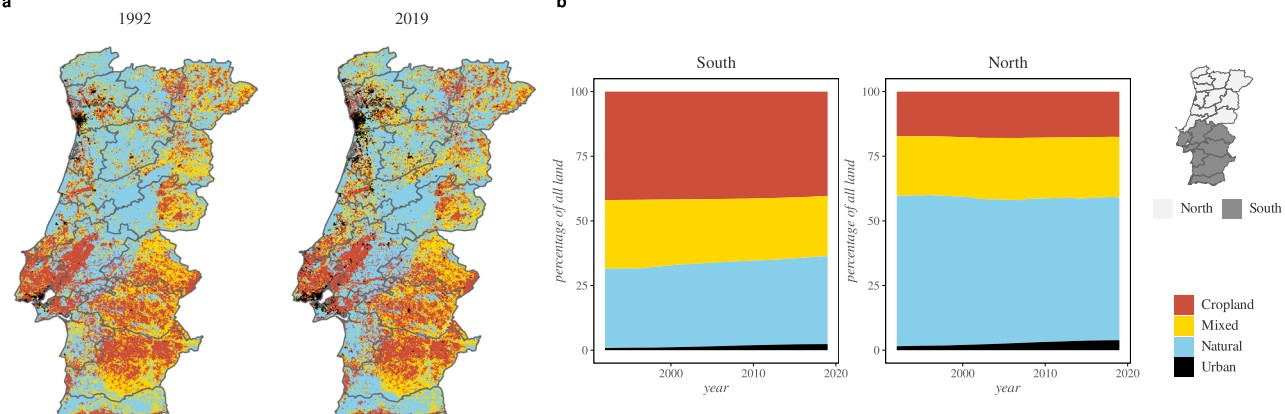

**Fig. 2 REVIVE Culex pipiens counts in Portugal (2017–2019).** Borders represent Portuguese counties, the level at which *Cx. pipiens* data are reported. **a**–**c** and **e**–**g** Counties are colored according to the yearly observations (transformed to $log_{10}(counts + 1)$) of adult and immature mosquitoes (respectively). **d**, **h** Counties are colored by the yearly mean observations (transformed to $log_{10}(mean(counts) + 1)$) of adult and immature mosquitoes (respectively). Counties for which there were zero counts are coloured in white. Counties for which collections were not performed are in grey.

**Fig. 3 Land types in Portugal (1992–2019). a** Shows the spatial distribution of different land types for the years 1992 (left) and 2019 (right). **b** is the percentage of land types over time for the south (left) and north (right) regions. In all panels, colours for land types are shown on the right. The North-South divide is the same as defined in the legend of Fig. 1 (boundaries shown on the map to the right).

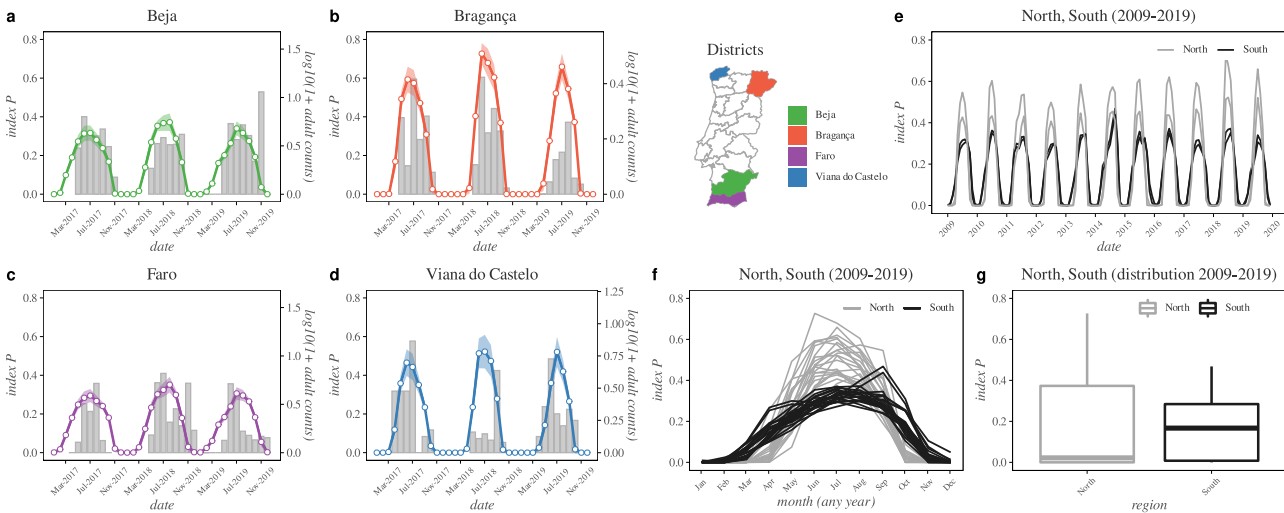

**Fig. 4 Monthly REVIVE Culex pipiens counts in Portugal versus estimated transmission potential index P (2017–2019). a–d** Present the monthly time series for $log_{10}(mosquito\ population\ size + 1)$ (grey bars) and mean (colored lines) and standard deviation (coloured areas) for four districts: two from the south (**a** for Beja in purple, **c** for Faro in purple) and two from the north (**b** for Bragança in red, **d** for Viana do Castelo in blue). The map panel presents all districts, coloured by their location. **e** Shows the mean index P for all districts 2009-2019 (north in grey and south in black). **f** Shows the seasonal index P per month from **e** independently of year. **g** is the distribution of index P across the period 2009–2019. For the estimation of the index P, the monthly mean of the climate variables was used per district. The North-South divide is the same as defined in the legend of Fig. 1.

both mosquito population size and estimated transmission potential. Seasons appeared to occur between the months of March and November of each year, often peaking in the summer and reaching minimum transmission potential values during the winter months. We calculated the peak timing for each region using the index P time series of all counties, and found that historically peaks occur in July, although on average 12 days later in the south compared to the north (t-test, $p < 0.01$, see **Analysis of peak suitability timing** for details). When comparing the northern and southern districts for a longer time period (2009–2019, Fig. 4e, f), we found that seasonal variation in index P typically presented higher values during the summer months in the north compared to the south ($\approx 35\%$ higher peaks, t-test $p < 0.001$). However, compared to the northern districts, the southern districts presented wider seasonal waves, with transmission potential increasing earlier in the spring and remaining non-zero well into late autumn (Fig. 4f). The higher values of P in the north, suggesting higher summer transmission potential compared to the south, contrasted with the existing evidence for WNV circulation (Fig. 1). However, the long-term time distribution of index P seasonal waves in the north was skewed, with lower medians than in the south (Fig. 4g), potentially supporting the North-South divide found in past evidence for WNV circulation in the country (Fig. 1).

To explore in detail the regional differences in transmission potential across the country, we next estimated the index P per available latitude-longitude coordinate in our climate dataset for the time period between 2016 and 2019 (Fig. 5). Yearly summary statistics per latitude-longitude coordinate revealed a North-South divide similar to the one presented by past evidence of WNV circulation in the country (Fig. 1). In particular, we found the southern region of Portugal to universally present higher median yearly index P when compared to the north (Fig. 5a, c, e, g). However, the geographical 'boundary' between north and south, in terms of WNV transmission potential, changed over the years (compare e.g. Fig. 5a, c). This output showed that, in accordance with previously described geo-temporal patterns for other mosquito-borne viruses[55] and WNV[54] in different countries, natural climate variations strongly influence both inter- and intra-yearly variation in transmission potential. When quantifying the number of months within a year for which the estimated transmission potential was not zero, we again found a clear division between the north and south

(Fig. 5b, d, f, h). Regions in the south and southwest typically spent nine or more months of the year with non-zero transmission potential, contrasting to regions in the north and northeast which spent only half or less of the year with non-zero transmission potential. Both the higher yearly median and longer seasonal waves in the south compared to the north, could thus be seen to support the mentioned North-South divide in past evidence for WNV circulation in Portugal (Fig. 1), hinting that the south may provide particularly favourable conditions for the virus.

Estimating transmission potential of a pathogen via the index P, while informed by climatic variables, provides means to reconstruct historical trends when long-term climatic data is available. We thus used the past 40 years of annual climate data per Portuguese county to explore long-time trends in climate variables and estimated WNV transmission potential, using linear quantile mixed models (see **Analysis of long-term climate and suitability trends** for technical details).

We found that all four explored variables presented significant differences between the north and south regions of the country (Fig. 6). Among these, precipitation had the smallest change over time, with a yearly decrease of $\approx -1.2 \times 10^{-6}$ in the north, and increase of $\approx 4.8 \times 10^{-6}$ in the south (Fig. 6a). Following the Köppen climate classifications, temperature was historically warmer in the south compared to the north. There were similar yearly increases of $\approx 0.041$ (north) and $\approx 0.039$ (south) Celsius per year between 1981 and 2019, equating to cumulative changes of $\approx 1.55$ and $\approx 1.44$ degrees for the north and south, respectively (Fig. 6b). As the country warmed, it also became dryer, with an accumulated decrease in relative humidity of $\approx 2.8\%$ in the north and $\approx 1.6\%$ in the south (Fig. 6c). Such historical climate trends translated into an yearly increase in estimated transmission potential for both regions (Fig. 6d). The index P had an yearly increase of $\approx 0.00097$, accumulating a change of $\approx 0.038$ between 1981 and 2019 in both regions.

As described in Fig. 4, the northern and southern regions presented an evident yearly periodicity in climate-driven transmission potential, with cycles being stable throughout the years and in phases between the regions. To explore the possibility of periodicity beyond the 12 month cycles, we performed wavelet analysis on the north and south yearly median

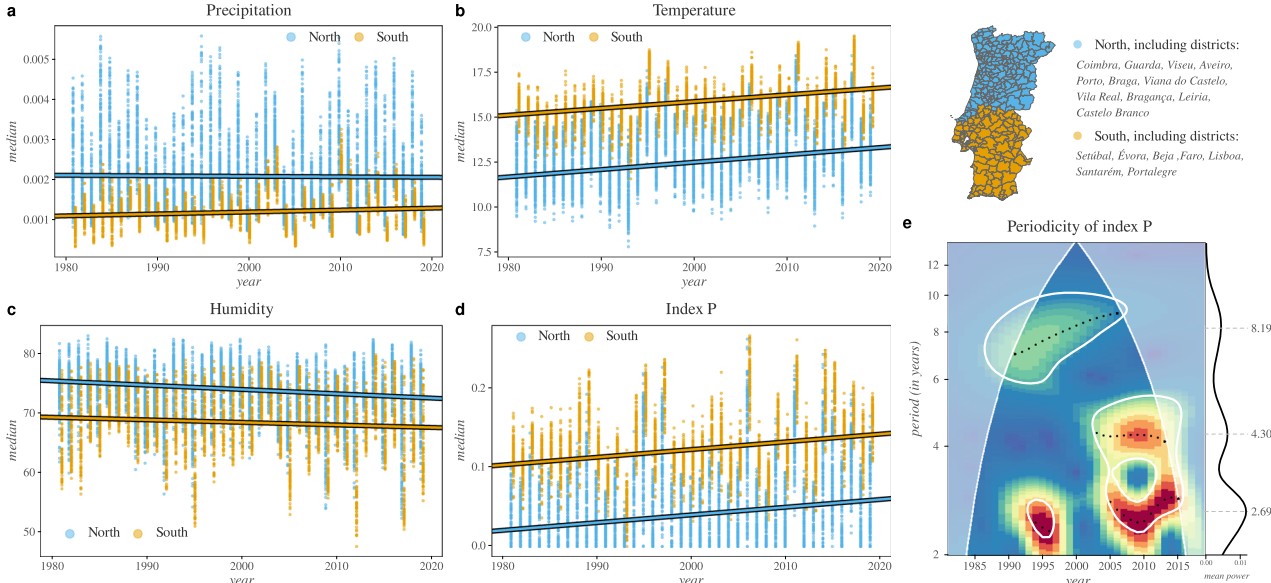

**Fig. 5 Estimated WNV transmission potential across Portugal (2016–2019).** Maps present summary statistics of estimated index *P* (WNV transmission potential) at the spatial resolution available in the climate dataset. **a**, **c**, **e** and **g** Present the yearly median index *P* per coordinate according to the presented color scale on the right. **b**, **d**, **f** and **h** Present the number of months during each year for which the estimated index *P* per coordinate was above zero, according to the presented color scale on the right. Boundaries in dark grey represent districts.

**Fig. 6 Long-term trends and periodicity of transmission potential of WNV and climate across different regions of Portugal.** **a**, **b** and **c** present the median of climatic variables among counties (points) within the north (blue) and south (orange) regions defined by the set of districts listed in the map subpanel (on the right). **d** Presents the same as **a**–**c** but for the index *P* (transmission potential). **e** Presents the cross-wavelet transform of the median yearly transmission potential between north and south time-series. Significant periodic components are circled in white (*p* − *value* < 0.01), where the heat map denotes power with peaks indicated by black dots (wavelet ridges) and mean power throughout time shown in the right subplot.

of transmission potential (see **Analysis of long-term suitability periodicity** for details). The analysis revealed longer periodic components in both regions, which were seen to change between 1981 and 2019 (Fig. 6e). Before 2005, periodicity was mostly found in ≈8 year cycles, although a strong 2–3 year component was present around 1995. From 2005 onward, periodicity shifted to become strongly dominated not by one but two components at 2–3 and 4–5 years. The presence of such quasi-periods indicates a Moran effect[57], i.e., that climate synchronizes transmission potential across large geographical areas in Portugal beyond the classic intra-year, summer-winter cycles. Finally, the significant shifts in the periodic components over the years do not support an historically stable influence of climate, but are rather suggestive that inter-year instability is an expected feature of WNV transmission potential in Portugal.

## Discussion

Recently, several European countries in the Mediterranean basin have progressed from sporadically reporting WNV infections to reporting yearly epidemic activity affecting both humans and equines (e.g., Spain, France, Greece[22]). To date, sufficient evidence has accumulated to support ongoing circulation of WNV in Portugal. However, despite Portugal sharing similar climate types and bird migration routes with nearby countries (e.g.,[58–60]) that have experienced WNV local dissemination, no events of sustained epidemic activity have been reported. Without such reports, the perception of the public health importance of WNV in Portugal remains minimal. Although several studies have focused on describing human, animal, and mosquito WNV infections in the country, its epidemiology remains largely uncharacterised and poorly understood. In this study, we have generated a historical, comprehensive perspective of the virus in Portugal using a combination of existing data and climate-informed theoretical approaches.

We collated existing data from the research literature, national databases, and recent serological samples in equines representative of the time period between 1966 and 2020. Recent data revealed that *Cx. pipiens*, considered to be the most suitable species for WNV transmission in Portugal, is present across the entire country. In contrast, we found that the vast majority of WNV-related evidence has been reported in the districts of Faro, Beja, Setúbal, Évora, Lisboa, Portalegre, and Santarém. This group of districts creates a clear North-South divide at the ≈39.5 latitude axis of the country, suggesting that past or ongoing WNV circulation is restricted to the south. Coincidently, such a divide is representative of the known geographical separation of the two Köppen climate classifications that exist in the country, with the north being classified as Temperate Mediterranean and the south as Warm Mediterranean.

Climate has long been accepted as a major driver of WNV epidemic activity and dissemination, given its effects on both the mosquito and avian species' population dynamics. To consolidate and complement existing data, we explored the effects of local climate variation on theoretical estimations of WNV transmission potential over the years. We estimated that, similarly to other countries in the northern hemisphere, WNV transmission seasons across the country tended to occur between March and November (Spring, Autumn), often peaking in July (Summer). The northern districts presented higher WNV transmission potential during the summer months, compatible with the expectation that WNV potential peaks at intermediate temperatures of 23–26 Celsius[19]. Indeed, during the summer months (June–August, 1981–2019), the average temperature of the northern districts was above 26 Celsius only ~3.1% of the time; as opposed to ~43% in the southern districts. Hence, the generally

warmer climate in the south may be detrimental for WNV transmission in the peak of the summer.

Additionally, we found that the inter- and intra-year median transmission potentials were substantially higher in the south when compared to the north. This implies that the south presents longer seasons suitable for WNV transmission, including the spring and autumn months; while the suitable seasons in the north are mostly restricted to the summer months. The importance of longer seasons suitable for WNV transmission rests on prior data: *Culex* mosquitoes from North America and Europe have been observed to shift their feeding preferences by late summer and early autumn, coinciding with a decrease in abundance of particular avian species and increases in reported infections[61,62]. Together, these differences between the north and the south of Portugal support the existing North-South divide in the evidence of WNV circulation.

In light of the ongoing discussion of the effects of climate change on the long-term epidemiology of mosquito-borne viruses, we explored historical climate variations and the corresponding WNV transmission potential in Portugal. Differences between the north and south of Portugal were clear and remained similar over the past 40 years. In general, we presented evidence that Portugal has slowly changed towards a warmer and dryer climate over this period. Temperature presented a especially large change across the years, with a median increase of +1 degree Celsius. Temperature remains a critical factor in mosquito-borne epidemiology, often positively affecting a large number of viral and entomological traits that favour transmission potential. Key examples are the positive relationship of higher temperatures with a shorter viral extrinsic incubation period, longer adult mosquito life-span, and shorter aquatic mosquito development[19,63,64]. Following the aforementioned historical trends in climate, we also found a small but continuous increase in transmission potential over the years. Our analyses also suggest that recent changes in climate, since 2005, appear to have introduced unstable inter-year variations in transmission potential, the practical consequences of which are difficult to assess.

Apart from climate, other ecological factors, such as the spatio-temporal distribution of land types, are known to affect the transmission potential of mosquito-borne viruses[48]. In Europe and North America, substantial positive associations have been found between human and animal WNV incidence and coverage of, or shorter distance to, cropland[49–53]. Associations between WNV incidence and natural types of land (e.g. forests, wetlands), or mixed land (cropland, natural), have also been reported[49,65–67]. Our results are consistent with these trends. We found that the south of Portugal is enriched with twice as much cropland compared to the north. Hence, it is possible that land type may also have contributed to the previously observed geographical distribution of WNV circulation in Portugal.

The data and modelling outputs presented include some limitations. For example, the geo-temporal patterns of transmission potential reflect solely the contribution of natural climate variation. Our estimations do not take into consideration other factors, such as the effects of land types and their proximity to human populations and animal farms, potential geo-temporal hotspots for inbound migratory avian species, mosquito and avian population sizes, biotype composition of the *Cx. pipiens* populations, etc. We emphasize that this study has focused on a suitability index (used previously for WNV in Israel) that does not include such factors, but rather focuses on climate alone. Furthermore, we note that precipitation is not included in the estimation of the suitability index. Many additional factors have been reported to be of potential interest for WNV epidemiology, and as such they can and should be the basis of future research, extending the methods and analyses presented in this study. Our theoretical

| Table 1 Studies including information on WNV or its vectors in Portugal. | | |
|---|---|---|
| **Group** | **PUBMED [in English]** | **Manual [in English & Portuguese]** |
| WNV in animals, in Portugal (serology/viral isolation/VNT) | 32,33,38,41,85 | 32,34,37,60 |
| WNV in vectors, in Portugal (viral isolation) | 35,39,40,85–88 | 37 |
| WNV in humans, in Portugal (serology/viral isolation/VNT) | 45,85 | 36,44 |
| WNV vectors in Portugal but no WNV research | 30,47,89–95 | 96 |
| WNV modelling including regions of Portugal | 97–99 | |
| WNV phylogenetics in Portugal | 88 | |

results are solely intended to summarize the direct effect of climate on WNV and should be interpreted as such. The original resolution of the climate data has also restricted some of our outputs. Namely, the resolution of our maps may have missed small-area regions of importance, and we were only able to estimate the duration of seasons (non-zero transmission potential) in the scale of months. The collated empirical data of past WNV circulation may be another source of limitations. For instance, some evidence was impossible to map geographically, given deficiencies in reporting (e.g. no region, or non-specific regions e.g. "South" or "Litoral"). It also remains unknown if the data includes a sampling bias towards the south of the country, which could partially explain the seemingly North-South divide in WNV circulation. This bias could arise from the reactive nature of surveillance initiatives responding to reports of symptomatic disease in sentinel species which may be distributed differently across the country, or alternatively, due to differences in WNV awareness among farmers, clinicians, and veterinarians.

The large European WNV epidemic of 2018 provided key opportunities to assess existing WNV surveillance and public health mechanisms. Countries implementing cross-sectoral and cross-disciplinary early warning systems based on a One Health surveillance approach (vector, avian, equine, and human) consistently reported positive impacts of such infrastructures[21]. Common areas for possible improvement included investment in and sustainability of mosquito surveillance, and improved information and media management. In Portugal, mosquito surveillance remains an exception in an otherwise fragmented and insufficient WNV surveillance infrastructure. Key improvements would include sustainability of laboratory capacities and their timely responses, active One Health surveillance beyond the mosquito level, and investment in awareness and training across public health domains and the general public, in particular in the south of Portugal.

## Conclusions

Our study provides a comprehensive, data-driven perspective of WNV in Portugal. Empirical data suggest a North-South discrepancy of WNV circulation in Portugal. It is still unclear whether WNV circulates endemically in any region of the country, but sufficient evidence exists to support ongoing seasonal circulation. Since data appeared unevenly geographically distributed across Portugal, we attempted to supplement them by describing distributions of mosquitoes and land types, as well as by estimating a theoretical WNV suitability index. The results support the geographical divide observed in molecular and serological data, and provide a first geo-temporal assessment of the contribution of local climate to the transmission potential of WNV in Portugal. The south of Portugal may be more suitable for WNV transmission due to effects of climate and land type, but the possibility of the north supporting transmission in shorter periods of each year can not be rejected. Furthermore, climate change is gradually increasing suitability for WNV in Portugal. A shift from passive to active One Health surveillance

will be necessary to manage future epidemics, which may result in public health emergencies, as reported recently in other European countries.

## Methods

**Literature reviews**. We queried PUBMED for articles related to WNV and Portugal using the query "((west nile virus) AND (Portugal)) OR (vírus nilo AND Portugal)" in all fields; thus leaving the query non-specific enough to capture a wide range of studies, and searching for articles both in Portuguese and English. The query returned 43 hits as of March 2021, from which 17 did not relate directly to the circulation of WNV or its vectors in the country. A second literature search was performed manually among Portuguese academic journals and PUBMED. References within the studies found in our two searches were also scanned for potential studies related to WNV circulation in Portugal. We also scanned the GenBank nucleotide database for existing WNV sequences from Portugal, searched with the query "((West Nile virus) AND (Portugal)) OR ((WNV) AND (Portugal))" OR (virus Nilo AND Portugal). The query returned 70 hits as of March 2021, of which 62 did not relate to WNV genomic regions.

We provide the results of our literature review in the form of tables. Studies found in the PUBMED and manual searches are grouped by content type in Table 1. All evidence related to the presence of WNV in Portugal across all studies found is summarized in Supplementary Table 1 (also available in a permanent figshare repository[68]). Studies presenting time series of relevant mosquito species in Portugal are summarized in Supplementary Table 2. All publicly available WNV genetic sequences obtained in Portugal are listed in Supplementary Table 3.

**Mosquito data**. We used mosquito data collected by the REVIVE network in Portugal[29]. Data included mosquito counts (adult and immature stages) for *Cx. pipiens* per county in Portugal during the period 2017–2019. The REVIVE data is in the public domain[29].

**Transmission suitability**. The basic reproduction number $R_0$ of WNV in the zoonotic reservoir can be summarised as the product of two factors: the number of adult female mosquitoes per host $M$, and the transmission potential of each adult female mosquito $P$ (i.e., $R_0 = MP$, Supplementary Fig. 1). Both $M$ and $P$ oscillate in time, driven by the influence of multiple eco-climatological factors. $M$, the ratio between the population size of mosquitoes $V$ and hosts $N$ ($M = V/N$), depends on the local effects of climate on mosquito survival and reproduction, affecting $V$[63], as well as on the reproduction and migratory behaviours of the avian species, affecting $N$[69]. For most epidemiological contexts and pathogens, high resolution data on either $V$ or $N$ is often lacking. The transmission potential $P$ depends on viral, mosquito and host factors (e.g. incubation periods, lifespans, etc), some of which depend on local climate variations[55,63,70] (Supplementary Fig. 1). Since adequate geo-temporal data for $V$ and $N$ is generally difficult to obtain, many recent theoretical epidemiology studies resort to estimating the transmission potential of mosquito-borne viruses using 'suitability indices'. These indices are functions of the parameters that compose either $M$ or $P$ or both, and generally depend on covariates such as climatic variables, vegetation indices, land types, host demography, mosquito and disease incidence reports, etc[55,71,72]. While a suitability index is an incomplete measure of the real transmission potential of a mosquito-borne virus, much work has been done to demonstrate the potential of an index to capture the geo-temporal dynamics of the number of reported mosquito-borne infections in host species[55,56,73].

In this study, we assessed local WNV transmission potential by estimating one of such suitability indices - termed the index $P$ (from $R_0 = MP$, Supplementary Fig. 1). The theory and practice of $P$ has been previously described in full by Obolski et al.[55]. The index $P$ has been used in several studies focusing on the population dynamics of the dengue, chikungunya, and Zika viruses (e.g.,[55,56,74,75]). Recently, the index was also successfully used to estimate the transmission potential of WNV in Israel dependent on local relative humidity and temperature variables[54]. In that study, $P$ was interpreted as a proxy for the risk of spillover to the human population given that it measured the WNV transmission potential by a single adult female mosquito in the animal reservoir. In the current study, we use the same approach as applied to Israel, informed by the epidemiological priors applied in that study (see Table in main text of[54]), which relate to *Culex* spp., WNV

**Table 2 Statistical output from the linear quantile mixed modelling.**

| Dependent variable | Year slope (95% CI) | South intercept (95%CI) | South and year interaction (95% CI): |
|---|---|---|---|
| Index P | $9.8 \times 10^{-4}$ ($9.4 \times 10^{-4}$, $10^{-3}$) | 0.082 (0.077, 0.088) | --------- |
| Humidity (relative, %) | $-0.07$ ($-0.08$, $-0.06$) | $-65$ ($-94$, $-36$) | 0.03 (0.015, 0.045) |
| Temperature (Celsius) | 0.04 (0.04, 0.04) | 9.6 (5.3, 13.9) | $-0.003$ ($-0.005$, $-0.001$) |
| Precipitation (m day-1) | $-1.2 \times 10^{-6}$ ($-1.8 \times 10^{-6}$, $-6.6 \times 10^{-7}$) | $-0.013$ ($-0.015$, $-0.014$) | $6 \times 10^{-6}$ ($5 \times 10^{-6}$, $7 \times 10^{-6}$) |

and a theoretical, average bird species. The choice of *Culex* spp. as the vector reflects its known ability to transmit WNV, together with evidence of its abundance in Portugal (see Section 3.2).

To inform the estimation of the index *P*, climatic data was downloaded from the Copernicus platform at www.copernicus.eu. We used the available dataset "Essential climate variables for assessment of climate variability"[76], which includes climatic variables at a time resolution of 1 month (1981–2019) and spatial resolution of 0.25°x0.25°. For each latitude-longitude location of the dataset, we used the MVSE R-package to estimate the index *P*[77] as described by Obolski et al. for Brazil, Honduras, and South America[55]. The results are available as a comma-separated file in a permanent figshare repository[68].

**Analysis of peak suitability timing and size.** Peaks in the monthly index *P* time series 1981–2019 per county were found by the maximal value and month where it occurred, using a circular mean to calculate the average time point of the year for the north and south regions, independently. The distribution of values throughout the years were then compared between regions with a circular ANOVA yielding a $p = 0.005$. Circular data was handled with the R-package 'circular'[78]. Mean peak values were calculated across counties for each year, using Welch's t-test to compare north and south regions yielding a $p = 1 \times 10^{-16}$.

**Analysis of long-term climatic and suitability trends.** We applied linear quantile mixed models to the estimated index *P*, temperature, precipitation, and humidity between 1981 and 2019, for which the Copernicus climate dataset was available. Our data contained yearly median values of the mentioned variables for each Portuguese county. Since the data included highly irregular points, which might skew standard linear regression models, we chose to use the more robust quantile regression using the median ($\tau = 0.5$). Furthermore, as different regions had repeated measures, we employed a mixed-effects quantile regression (using the 'lqmm' R-package[79]). Our models contained a random intercept term for each region, a fixed effect dummy variable coding for north versus south, and a continuous variable representing the time unit (year). We tested the interaction between the year and North-South terms and found it statistically significant ($p<0.01$) in all models except when index *P* was the dependent variable ($p > 0.7$) for which the term was dropped. All the other fixed effects were statistically significant for all models ($p<0.001$) and are summarized in Table 2.

**Analysis of long-term suitability periodicity.** Detrending and wavelet analysis followed the workflow of Damineli et al.[80]. Detrending with a Maximal Overlap Discrete Wavelet Transform was performed in a multiresolution analysis with the R-package 'wavelets'[81]. Significant periods of the continuous wavelets were assessed considering the spectrum of an autoregressive model of order 1, as a null model using the R-package 'biwavelet'[82].

**Equine WNV surveillance data 2016–2020.** Between 2016 and 2020 the virology laboratory of INIAV performed molecular and serology testing in 165 horses from several regions of the country, reacting to local suspected neuroinvasive disease. RNA was extracted by using the BioSprint 96 workstation with the MagAttract 96 cador pathogen kit, according to manufacturer's instructions (Qiagen, Hilden, Germany). The samples were screened for WNV by real-time reverse transcriptase PCR (RT-qPCR) targeting the NS2A gene[83]. Equine sera were tested for WNV-IgM antibodies by capture ELISA (ID Screen West Nile IgM Capture ELISA, IDVET, Montepellier, France). Twenty-three horses were diagnosed as WNV-positive on the basis of IgM positivity, with one horse being also positive by RT-qPCR (July 2020, Escalos de Cima-Castelo Branco, center Portugal). Four of eight horses that were followed for clinical progression died or were euthanized. These results are summarized in Supplementary Table 1, including the time and region of sampling.

**Land cover and land types.** Land data were downloaded from the Copernicus platform at www.copernicus.eu. We used the available dataset "Land cover classification gridded maps from 1992 to present derived from satellite observations"[84], which includes land surface classification with 22 classes at a time

resolution of 1 year (1992–2019) and a spatial resolution of 300 m. Land cover classification were aggregated into four classes: cropland vegetation (classes 10, 11, 12, and 20), natural vegetation (classes 60, 70, 90, 100, 110, 120, 122, 130, and 180), mixed vegetation (i.e., natural and cropland; classes 30, 40) and urban (class 190).

**Reporting summary.** Further information on research design is available in the Nature Research Reporting Summary linked to this article.

## Data availability

All datasets generated and analysed during the current study are publicly available, including: compiled evidence of WNV circulation in Portugal (figshare repository[68], also in Tables and Supplementary Tables), REVIVE mosquito data (in yearly official reports[29]), climate data (Copernicus: "Essential climate variables for assessment of climate variability"[76]), land cover data (Copernicus: "Land cover classification gridded maps from 1992 to present derived from satellite observations"[84]). Estimations of index P generated in this study are also made available on figshare[68].

## Code availability

We used the standard scripts of MVSE R-package version 1.01r for the estimations of the transmission potential (code snapshot of version 1.01r available on Figshare: https://doi.org/10.6084/m9.figshare.17026331.v1). Code examples are available in the package's documentation[77] and manuscript[55].

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

## Acknowledgements

We are grateful to The Ministry of Health / National Institute of Health (INSA) under the National Vector Surveillance Network - REVIVE for supporting this research. We are also grateful to the REVIVE team for the mosquito collection nationwide, and to Patrícia Rosa Ramos Rodrigues for the collection of the reported CSF equine sample. DSCD was funded by the grant 19/23343-7 and 20/06160-3 within the scope of 15/22308-2 from the São Paulo Research Foundation (FAPESP). JL was supported by a research lectureship by the Department of Zoology, University of Oxford. MG was supported by Fundação de Amparo à Pesquisa do Estado do Rio de Janeiro (FAPERJ). Funding sources had no involvement in the design and interpretation of the presented research.

## Author contributions

J.L.: Data Curation, Resources, Methodology, Conceptualization, Software, Formal analysis, Investigation, Writing - Original Draft, Writing - Review & Editing, and Supervision. S.C.B: Data Curation, Methodology, Writing - Original Draft, Writing - Review & Editing, and Supervision. L.Z.Z.: Data Curation, Resources, Methodology, Conceptualization, Formal analysis, Writing - Original Draft, Writing - Review & Editing, and Supervision. D.S.C.D: Data Curation, Methodology, Software, Formal analysis. Marta Giovanetti: Data Curation, Methodology, Software, Writing - Original Draft, Writing - Review & Editing. H.C.O.: Data Curation, Writing - Review & Editing. F.A.: Data Curation, Resources. A.M.He.: Data Curation, Resources. F.R.: Data Curation, Resources. T.L.: Data Curation, Resources. M.D.D.: Data Curation, Resources. T.F.: Data Curation, Resources. M.J.A.: Data Curation, Resources, Writing - Review & Editing, Supervision. U.O.: Data Curation, Resources, Methodology, Conceptualization, Software, Formal analysis, Investigation, Writing - Original Draft, Writing - Review & Editing, Supervision.

## Competing interests

The authors declare no competing interests.
