## [Peer Review File · Communications Biology]

Reviewers' comments:

Reviewer #1 (Remarks to the Author):

In this manuscript the authors aimed to provide a first broad overview of WNV dynamics in Portugal over the last 50 years. The authors first collated the sparse data available about WNV occurrence in Portugal. The few evidences provided by data were then supported by a climate-driven index which helped to shed some more light on the transmission dynamics of WNV virus in Portugal. By using this data and model driven approach, they were able to find and explain some interesting trends such as a WNV North-South "divide" in the country. The manuscript represents an important starting point upon which other studies can build to better characterise WNV dynamics in Portugal at finer temporal, spatial and epidemiological scales.

I've only a few comments:

While the role of temperature in WNV dynamics is out of discussion, other environmental factors such as precipitation (droughts) have been often indicated as possible correlated of changes in WNV dynamics (also if the slope of the change is minimal as in your study). The authors analyse precipitation (which is integrated in the P index) however they fail to discuss its potential role in WNV dynamics. It would be important to have a paragraph including the role of precipitation in the discussion.

Moreover, the paper lacks an analytic consideration of the effect of land use/ land use changes on WNV, while these are important drivers for the emergence of mosquito-borne viruses such as WNV. The authors transparently pointed out this limitation in the discussion and I appreciated this. However, they state that they did not consider land use due to the lack of available data at "adequate resolutions". This sounds odd as Copernicus programme (from which climatic data were taken) offers many datasets at spatial resolution often finer than climatic data, such as CLC or CCI datasets (other datasets are available). Other programmes, such as The MODIS Terra and Aqua Combined Land Cover product offer similar datasets.

Minor comments:

The draft does not have line numbering so I'm indicating the page:

Page 1: The species of mosquitoes that contributes to the WNV cycle are not so many, please add numbers to support your statement or rephrase it.

Page 14: Remove capital from "Hence"

Reviewer #2 (Remarks to the Author):

I found this study very interesting to read. The authors compile an important summary of all available knowledge on WNV circulation in Portugal. They also carried out some modelling efforts offering compelling evidence of the country suitability for WNV transmission. Portugal indeed represents so far a peculiar case with respect to the rest of Europe regarding WNV transmission, and I agree with the authors when they call for an improvement in WNV surveillance in the country. I have some minor comments and suggestions for the authors, which I list below.

1. Page 7. "many local clinicians and veterinarians lack awareness". I agree with the authors this might be a very important issue, however please provide some reference to substantiate this claim
2. Page 4, section 2.2. So is *Culex pipiens* the most likely potential vector in Portugal?
3. Page 4, section 2.3. Please explain briefly which parameters of index P are affected by temperature and how. From reference 36 it seems that the functions were designed for *Aedes aegypti* mosquitoes. Was then everything adapted to *Culex* mosquitoes?
4. Page 8. Can you provide a full list of avian and mosquito species found to be WNV positive? Or are the few species reported here within brackets a complete inventory?

4. Figure 3. Panels A-D: color lines are for index P, right? Please use the same scale for the y axes so that it is easier to compare the four districts.

6. Page 13. It is not very clear to me how the absolute difference of 0.083 was computed. Was it a sum of the yearly differences?

7. Page 13. Bird migration routes: please provide some reference.

8. Did your literature search yield any study failing to find WNV? For instance, serological surveys finding no antibodies or screening in mosquitoes with no virus isolation. If so, how did you consider them? If you found only positive studies, this might be discussed as a bias (i.e., studies were carried out probably already knowing the virus was there).

Dear reviewers and editor,

Please find below all of your comments (in bold, dark blue) with a point-by-point response (regular font). We have answered all the comments raised and have followed your suggestions, making significant changes to the manuscript. A version of the manuscript is submitted with the changes highlighted in yellow. We hope that you now find the manuscript suitable for publication.

Yours sincerely, on behalf of all authors, the corresponding author.

Reviewer #1 (Remarks to the Author):

In this manuscript the authors aimed to provide a first broad overview of WNV dynamics in Portugal over the last 50 years. The authors first collated the sparse data available about WNV occurrence in Portugal. The few evidences provided by data were then supported by a climate-driven index which helped to shed some more light on the transmission dynamics of WNV virus in Portugal. By using this data and model driven approach, they were able to find and explain some interesting trends such as a WNV North-South “divide” in the country.

The manuscript represents an important starting point upon which other studies can build to better characterise WNV dynamics in Portugal at finer temporal, spatial and epidemiological scales.

Thank you for the kind words and for taking the time to review our manuscript.

I've only a few comments:

While the role of temperature in WNV dynamics is out of discussion, other environmental factors such as precipitation (droughts) have been often indicated as possible correlated of changes in WNV dynamics (also if the slope of the change is minimal as in your study). The authors analyse precipitation (which is integrated in the P index) however they fail to discuss its potential role in WNV dynamics. It would be important to have a paragraph including the role of precipitation in the discussion.

We apologize for not being sufficiently clear in the original text. Our approach is the same as the one used to study WNV in Israel - that is, the Index P is not informed by precipitation in the estimation process. We had mentioned precipitation as input for the R-package that we use to estimate the index P, since other functions in the same R-package use it; but it wasn't our intention to associate precipitation with P in the text. Precipitation was only examined as an additional climatic variable over the years (Figure 6) for completeness, and given its possible relationship with humidity and temperature. We have now added a Supplementary Figure explicitly describing the role of each climatic variable in the calculation of index P, and clarify in the text that precipitation, among other variables, is not used in our suitability estimations (Discussion):

“We emphasize that this study has focused on a suitability index (used previously for WNV in Israel) that does not include such factors, but rather focuses on climate alone. Furthermore, we note that precipitation is not included in the estimation of the suitability index. Many additional

factors have been reported to be of potential interest for WNV epidemiology, and as such they can and should be the basis of future research, extending the methods and analyses presented in this study. Our theoretical results are solely intended to summarize the direct effect of climate on WNV and should be interpreted as such."

Moreover, the paper lacks an analytic consideration of the effect of land use/ land use changes on WNV, while these are important drivers for the emergence of mosquito-borne viruses such as WNV. The authors transparently pointed out this limitation in the discussion and I appreciated this. However, they state that they did not consider land use due to the lack of available data at "adequate resolutions". This sounds odd as Copernicus programme (from which climatic data were taken) offers many datasets at spatial resolution often finer than climatic data, such as CLC or CCI datasets (other datasets are available). Other programmes, such as The MODIS Terra and Aqua Combined Land Cover product offer similar datasets.

We thank the reviewer for noticing this detail, as indeed land use / type data is available from Copernicus. We have now added a substantial new section to the Results, that includes a new figure with both spatial land type distributions and their temporal trends (3.3 *Land types in Portugal*):

"The distribution of land types (e.g. cropland, urban, natural) can modulate the transmission of mosquito-borne pathogens and the risk for cross-species transmission [69,70]. Therefore we used the distribution of different land types in Portugal (see 2.8 Land cover and land types for details) to explore how they could contribute to the explanation of the North-South dichotomy in WNV circulation data.

We mapped the spatial distribution of land types (Figure 3A) and their frequencies between 1992 and 2019 (Figure 3B). The percentage of mixed land (natural, cropland) was similar between the regions (~23% of all land). The South was enriched in cropland (~40%), approximately twice as much as in the North. In contrast, the North was dominated by natural vegetation (~55%), approximately double of what was observed in the South. The spatial distribution of different land types, which was quite stable over time, highlighted differences between the regions that were reminiscent of the North-South divide in existing WNV data. The abundance of cropland in the South appeared linked with reports of WNV circulation, which conforms to mounting evidence of the positive link between cropland and the transmission activity of WNV both in humans and animals [70–74]."

Moreover, we address this in the revised Discussion:

"Apart from climate, other ecological factors, such as the spatio-temporal distribution of land types, are known to affect the transmission potential of mosquito-borne viruses [69]. In Europe and North America, substantial positive associations have been found between human and animal WNV incidence and coverage of, or shorter distance to, cropland [70–74]. Associations between WNV incidence and natural types of land (e.g. forests, wetlands), or mixed land (cropland, natural), have also been reported [70,82–84]. Our results are consistent with these trends. We found that the south of Portugal is enriched with twice as much cropland compared to the north. Hence, it is possible that land type may also have contributed to the past observed geographical distribution of WNV circulation in Portugal."

Minor comments: The draft does not have line numbering so I'm indicating the page:

Page 1: The species of mosquitoes that contributes to the WNV cycle are not so many, please add numbers to support your statement or rephrase it.

Indeed, WNV has been detected in a huge number of mosquito species (some references indicate 65 species or even more, and worldwide 6 genera: *Anopheles*, *Aedes*, *Culex*, *Culiseta*, *Mansonia* and *Ochlerotatus*). The *Culex* species most commonly referred to as vectors of WNV in Europe are *Cx. pipiens* and *Cx. univitattus/perexiguus* (species complex). *Cx. modestus* is also sometimes discussed. To clarify, we have changed the sentence in the introduction to:

“WNV ecology is characterised by a zoonotic transmission cycle maintained between mosquitoes and wild avian species. Six mosquito genera have been implicated in WNV transmission (*Anopheles*, *Aedes*, *Culex*, *Culiseta*, *Mansonia*, *Ochlerotatus*), but *Culex spp.* (e.g. *Cx. pipiens*, *Cx. univitattus / perexiguus*, *Cx. modestus*) are commonly referred to as the main vectors of WNV in Europe and North America.”

Page 14: Remove capital from “Hence”

We had missed a period at the end of the sentence. This has been fixed.

Reviewer #2 (Remarks to the Author):

I found this study very interesting to read. The authors compile an important summary of all available knowledge on WNV circulation in Portugal. They also carried out some modelling efforts offering compelling evidence of the country suitability for WNV transmission. Portugal indeed represents so far a peculiar case with respect to the rest of Europe regarding WNV transmission, and I agree with the authors when they call for an improvement in WNV surveillance in the country. I have some minor comments and suggestions for the authors, which I list below.

Thank you for the kind words and for taking the time to review our manuscript.

Page 7. "many local clinicians and veterinarians lack awareness". I agree with the authors this might be a very important issue, however please provide some reference to substantiate this claim

We are not aware of any reference to substantiate this sentence. This statement comes from daily knowledge of the Portuguese authors who are involved in WNV passive surveillance in the country. This experience regards the human diagnostic requests for neurotropic Arbovirus to the national reference laboratory which sums to (on average) about 50 a year. We are aware that the number of requests increases substantially when human cases are reported in the country (and also a bit, but less frequently associated with positive cases in horses) and in neighboring Spanish regions. We also know that the number of requests for non-vector borne neurotropic viruses is about ten times higher, and a significant percentage of these diagnostic requests is negative (historically remaining without the identification of the aetiological viral agent).

To make this clear, we have changed the sentence in the manuscript to:

“Furthermore, according to the experience of the national reference laboratory for vector-borne viruses, practitioners in Portugal lack awareness of the importance of WNV screening in compatible disease cases.”

Page 4, section 2.2. So is *Culex pipiens* the most likely potential vector in Portugal?

Without the occurrence of regular epidemics to quantify associations with mosquito species, or active surveillance able to detect carrying mosquitoes more frequently, it is really impossible to quantify the relative role of each *Culex* species. However, we argue that *Culex pipiens* is indeed a likely WNV vector in Portugal, since it has been previously associated with 2 human cases detected in 2004 in the Algarve. It is also the species for which we have substantially more data about, making it possible to explore results such as the ones in Figures 2 and 4. We thus agree that this is a relevant question, but one that cannot be fully answered at the moment, and indeed one that may be answered in the near future if surveillance is changed towards an active system, which we advocate for in the manuscript. This is now explained in the text (Methods):

“The choice of *Culex* spp. as the vector reflects its known ability to transmit WNV, together with evidence of its abundance in Portugal (see Section 3.2).”

Page 4, section 2.3. Please explain briefly which parameters of index P are affected by temperature and how. From reference 36 it seems that the functions were designed for *Aedes aegypti* mosquitoes. Was then everything adapted to *Culex* mosquitoes?

We acknowledge that this was inadequately communicated in the original manuscript. We have now made changes to the text (Methods) and have added Supplementary Figure S1 to correct it. The latter presents the mathematical expression of the index P and identifies all of the parameters that are climate dependent and on which climatic variables they depend. We believe these changes are sufficient to avoid readers having to check the material of our previous studies.

Regarding the functions and *Ae. aegypti* - yes, the original formulation was based on this species of mosquito, for which vast and high resolution empirical data exist, to be able to robustly quantify the relationships of each entomological / viral parameter and climatic variables. We have previously demonstrated, using epidemiological time series of WNV incidence in Israel, that transferring those (*Ae.*) relationships to WNV is highly informative; and we apply this same strategy in the current manuscript. However, it should be noted that, where possible, we used priors for parameters that pertain to *Culex* spp.. We adapted the Methods text to be more explicit about this:

“Recently, the index was also successfully used to estimate the transmission potential of WNV in Israel dependent on local relative humidity and temperature variables [43]. In that study, P was interpreted as a proxy for the risk of spillover to the human population given that it measured the WNV transmission potential by a single adult female mosquito in the animal reservoir. In the current study, we use the same approach as applied to Israel, informed by the epidemiological priors applied in that study (see Table in main text of [43]), which relate to Culex spp., WNV and a theoretical, average bird species. The choice of Culex spp. as the vector reflects its known ability to transmit WNV, together with evidence of its abundance in Portugal (see Section 3.2).”

Page 8. Can you provide a full list of avian and mosquito species found to be WNV positive? Or are the few species reported here within brackets a complete inventory?

We have now added a full description of avian and mosquito species found positive for WNV to Table 2.

Figure 3. Panels A-D: color lines are for index P, right? Please use the same scale for the y axes so that it is easier to compare the four districts.

We agree that this would improve the figure, and have followed the suggestion. The figure has now been changed. Please note that old Figure 3 is now Figure 4.

Page 13. It is not very clear to me how the absolute difference of 0.083 was computed. Was it a sum of the yearly differences?

The reviewer's question relates to the sentence "*During this period, the South had an absolute difference of 0.083 in estimated transmission potential compared to the North*". We thank the reviewer for pointing this out, because on revision we acknowledge how non-informative this sentence is. The value 0.083 was the mean difference between the mean of the north and the mean of the south per year. As noted in Figure 6D's regressions, this difference was virtually constant over the years - this was what we intended to communicate in the original version of the manuscript. We have now removed the noted sentence, and changed the end of the previous sentence to make our point clearer:

"The index P had an yearly increase of ≈ 0.00097 , accumulating a change of ≈ 0.038 between 1981 and 2019 in both regions."

Page 13. Bird migration routes: please provide some reference.

This was indeed missing. We note that a major European route for intercontinental bird migration exists over Portugal - the East Atlantic flyway. We have now added three references to support this fact (including the previously cited reference 73 that reported mosquito-borne pathogens in migratory birds in Portugal).

Did your literature search yield any study failing to find WNV? For instance, serological surveys finding no antibodies or screening in mosquitoes with no virus isolation. If so, how did you consider them? If you found only positive studies, this might be discussed as a bias (i.e., studies were carried out probably already knowing the virus was there).

We agree with the reviewer that this is an important point. When we performed the literature search, there were no restrictions on whether studies had positive or negative results. As described in the Methods, our search included all studies that related Portugal to WNV (in fact, even if the studies were not based in Portugal, but presented reference to WNV related topics in Portugal). The studies included in Table 1 thus include both positive and negative results. In

contrast, Table 2 describes in detail which studies of Table 1 have presented / described / found positive results.

REVIEWERS' COMMENTS:

Reviewer #1 (Remarks to the Author):

The authors replied all my comments and edited the draft accordingly. I don't have any additional suggestions.

Reviewer #2 (Remarks to the Author):

I think the authors addressed fairly and thoroughly all my previous comments.

I've a very small last remark. I noticed that the last cited item (ref 99, Spring temperature shapes West Nile virus transmission in Europe) lacks the authors.